# Wastewater sequencing reveals community and variant dynamics of the collective human virome

Michael Tisza [1,2,11], Sara Javornik Cregeen[1,2,11], Vasanthi Avadhanula[2], Ping Zhang[1,2], Tulin Ayvaz[1,2], Karen Feliz[2], Kristi L. Hoffman[1,2], Justin R. Clark[2,3], Austen Terwilliger[2,3], Matthew C. Ross [1,2], Juwan Cormier[1,2], Hannah Moreno[1,2], Li Wang[1,2], Katelyn Payne[1,2], David Henke [2], Catherine Troisi[4], Fuqing Wu[4,5,6], Janelle Rios[4,5], Jennifer Deegan[4,5], Blake Hansen [4,5,6], John Balliew[7], Anna Gitter[4,5,6], Kehe Zhang [4,8,9], Runze Li[4,8,9], Cici X. Bauer [4,5,8,9], Kristina D. Mena[4,5,6], Pedro A. Piedra [2,10], Joseph F. Petrosino[1,2] ✉, Eric Boerwinkle[4,5,6] ✉ & Anthony W. Maresso[2,3] ✉

Wastewater is a discarded human by-product, but its analysis may help us understand the health of populations. Epidemiologists first analyzed wastewater to track outbreaks of poliovirus decades ago, but so-called wastewater-based epidemiology was reinvigorated to monitor SARS-CoV-2 levels while bypassing the difficulties and pit falls of individual testing. Current approaches overlook the activity of most human viruses and preclude a deeper understanding of human virome community dynamics. Here, we conduct a comprehensive sequencing-based analysis of 363 longitudinal wastewater samples from ten distinct sites in two major cities. Critical to detection is the use of a viral probe capture set targeting thousands of viral species or variants. Over 450 distinct pathogenic viruses from 28 viral families are observed, most of which have never been detected in such samples. Sequencing reads of established pathogens and emerging viruses correlate to clinical data sets of SARS-CoV-2, influenza virus, and monkeypox viruses, outlining the public health utility of this approach. Viral communities are tightly organized by space and time. Finally, the most abundant human viruses yield sequence variant information consistent with regional spread and evolution. We reveal the viral landscape of human wastewater and its potential to improve our understanding of outbreaks, transmission, and its effects on overall population health.

Wastewater-based epidemiology (WBE) refers to the specific detection and tracking of substances[1], chemicals[2], genes[3,4], or pathogens[5] in municipal sewage or sludge to assess population health or disease risk. During the COVID-19 pandemic, the WBE field underwent significant reinvestment[6], wherein PCR-based detection of SARS-CoV-2 was used as a proxy for community infection levels, and amplicon sequencing facilitated the resolution of SARS-CoV-2 variants well before clinical detection[7–9]. As such, viral WBE, while initially used for environmental poliovirus surveillance nearly a century ago[10], has now been leveraged to track influenza virus[11], respiratory syncytial virus[12], enterovirus

D68[13], and monkeypox virus[14,15] using modern PCR-based methods. Although delivering high sensitivity and specificity, these methods are limited as they cannot provide a comprehensive assessment of human virus levels, community diversity, and variant compositions in a heterogeneous sample.

Recent approaches have investigated the use of virus-like particle enrichment[16], targeted amplification[8,9], and/or hybrid capture methods[17–20] to enrich for rare viral sequences amongst the backdrop of what is mostly nucleic acid from bacteria and mammalian hosts. These studies have seen mixed success, not been applied at scale, and the extent to which the levels of virus sequences corresponded to community infection levels is unclear. In any case, even the most prevalent wastewater viruses, e.g., human astroviruses and rotaviruses, comprise a very small fraction of the total biomatter in wastewater, especially compared to other microorganisms, such as bacteria[4,17,21,22].

Clinical reporting of infectious diseases is extremely valuable for understanding potential sources of outbreaks and disease burden on those with co-morbidities and certain population demographics, but it is constrained by resources, changes in human behavior, and trends in clinical practice. We demonstrate that WBE employing virome sequencing provides insights into aggregate community loads of specific pathogens, viral evolution, dynamics between different viral species or variants, and is presumably agnostic to clinical reporting biases. Specifically, we apply a probe-based capture method accounting for thousands of human and animal viruses followed by deep sequencing to wastewater samples from two major cities whose combined populations reach nearly 3 million people. We reveal the dynamics of the human virome in space and time from hundreds of pathogenic viruses, correlate some of this activity to established detection platforms and clinical data sets, and identify widespread allelic variants of specific viruses for evolutionary tracking.

## Results

### Probe-based capture drives viral enrichment

We developed a comprehensive viral capture approach using a diverse probe set across ten different sites on a weekly basis for nearly 1 year. The probes (TWIST Comprehensive Viral Capture Panel) are directed against a panel of 3153 different human and animal virus genomes. As part of an initiative from the Texas Epidemic Public Health Institute[23], composite 24-h wastewater influent was collected from six treatment plants in Houston, Texas, USA and four plants El Paso, Texas, USA from May 2022 through February 2023 (Fig. 1A). Wastewater treatment plant catchment areas varied between 10,000 and 400,000 people (estimated 618,148 people served in Houston and 751,982 in El Paso County). These sites were chosen because they allowed us to examine the breadth and robustness of our approach across two large cities with different characteristics. Houston and El Paso also differ in size and diversity, have contrasting climate and rainfall (El Paso dry and Houston humid), are geographically distant (almost 1200 kilometers), and have different patterns of human travel (El Paso a border city with thousands of daily cross-border commuters, Houston a coastal city with one of the largest ports in the world).

The efficacy of probe-based enrichment methods was tested on 18 pilot samples. Following clearance of solids and nucleic acid extraction using methods designed for SARS-CoV-2 detection[24], we first sequenced and examined viral read numbers from unenriched samples. Low proportions of viral reads were derived from these unenriched samples (4 – 78 aligned reads out of 9.8 – 18.0 million total reads), with 0 to 1 total mammalian viruses detected. In contrast, utilizing the TWIST Comprehensive Viral Research Panel probes on the same extractions, a 3,374-fold enrichment in the proportion of virus reads was observed (Fig. 1F) (14.9 thousand – 407.0 thousand aligned reads out of 11.6 – 24.2 million total reads), with 42 to 128 total mammalian viruses detected).

Read mapping-based virus detection and abundance measurement was conducted using EsViritu, a bioinformatics tool we developed for this purpose (Fig. S1). EsViritu leverages sequence information to sensitively detect mammalian viruses and filter out false positives (see materials and methods) (https://github.com/cmmr/EsViritu).

Applying these methods to 363 longitudinal wastewater samples, we detected 28 viral families, 77 genera, 191 species, and 465 distinct virus strains in total (Fig. 1B), with a median of 54 to 98 strains detected per sample, depending on the wastewater treatment plant (Fig. 1C). Furthermore, rarefaction analysis of virus strains showed that the unique detections were not saturated, and additional virus strains are likely to be detected in future samples (Fig. 1D). A median of 28.5 reference genomes or segments had sequencing reads aligning to over 90% of their length with an additional 41 (median) genomes or segments with over 50% alignment (Fig. 1E). From a methodological standpoint, this emphasizes the potential for in-depth analysis of circulating viruses beyond abundance measurements.

To infer the quantitative dynamic range for pathogen detection of this assay, we added in lab-grown respiratory syncytial virus A (RSV) virions to real wastewater samples (samples were previously determined to have no detectable RSV). Based on a stepwise dilution series, we could accurately detect and quantify RSV from a spike-in of 51 genome copies to 4 million genome copies with a Pearson correlation of at least $R = 0.975$ (Fig. S2).

### Correlation of viral sequencing data with clinical cases

Having established a capture-based approach that offers the prospect of a comprehensive virome analysis of complex wastewater samples, we next asked whether signals generated from sequencing data mirror trends observed from publicly available clinical datasets. Case data from select viral pathogens, namely SARS-CoV-2, influenza virus, and monkeypox virus, were obtained for Houston and, when available El Paso, from local or state government sources. We started first with SARS-CoV-2, as wastewater levels have previously been correlated with case data[25]. Using the reads per kilobase of transcript per million filtered reads (RPKMF) as a proxy for relative virus levels in a given sample, there was a positive correlation between case data and positivity rate for SARS-CoV-2 summer and winter waves and the wastewater signal in Houston (Fig. 2A, S3A, B, $R = 0.5$–0.78) and case data from El Paso (Fig. 2B, $R = 0.59 - 0.73$). This finding is strengthened by the fact that a second orthogonal technique to measure SARS-CoV-2 levels in wastewater (i.e., qPCR which is the current standard) was also closely correlated with the RPKMF (Fig. S3C, D) for both Houston ($R = 0.64$) and El Paso ($R = 0.84$).

Similarly, Influenza A Virus abundance in the virome sequencing data was highly concordant with reporting of "Weekly Percentage of Visits with Discharge Diagnosed Influenza" in the Houston area (Fig. 2C, $R = 0.9$). Influenza variants H3N2 and H1N1 were also resolved in our data, concordant with clinical subtyping of this flu season in Texas (see Data and Materials Availability). Once more, the virome sequencing data was highly correlated with qPCR measurements from the same samples (Fig. S3E, F, $R = 0.57 - 0.73$). Finally, a Monkeypox (Mpox) outbreak occurred in the summer of 2022 in several U.S. cities. Rather strikingly, monkeypox virus was detected numerous times at low abundance in Houston wastewater samples (Fig. 2D, $R = 0.46$) in our virome dataset, even though only 1,050 cases were reported in the entire Houston area between July and November 2022. Meanwhile, no detection events of monkeypox virus were recorded from El Paso wastewater samples, consistent with only 10 total reported clinical cases in this metro area.

Encouraging from a detection and possibly public health standpoint, 11 categories of "major" viral pathogens were routinely detected and could be tracked over the sampling period (Fig. 2E), including noroviruses, rotavirus A, hepatitis A virus, RSV, parainfluenza viruses,

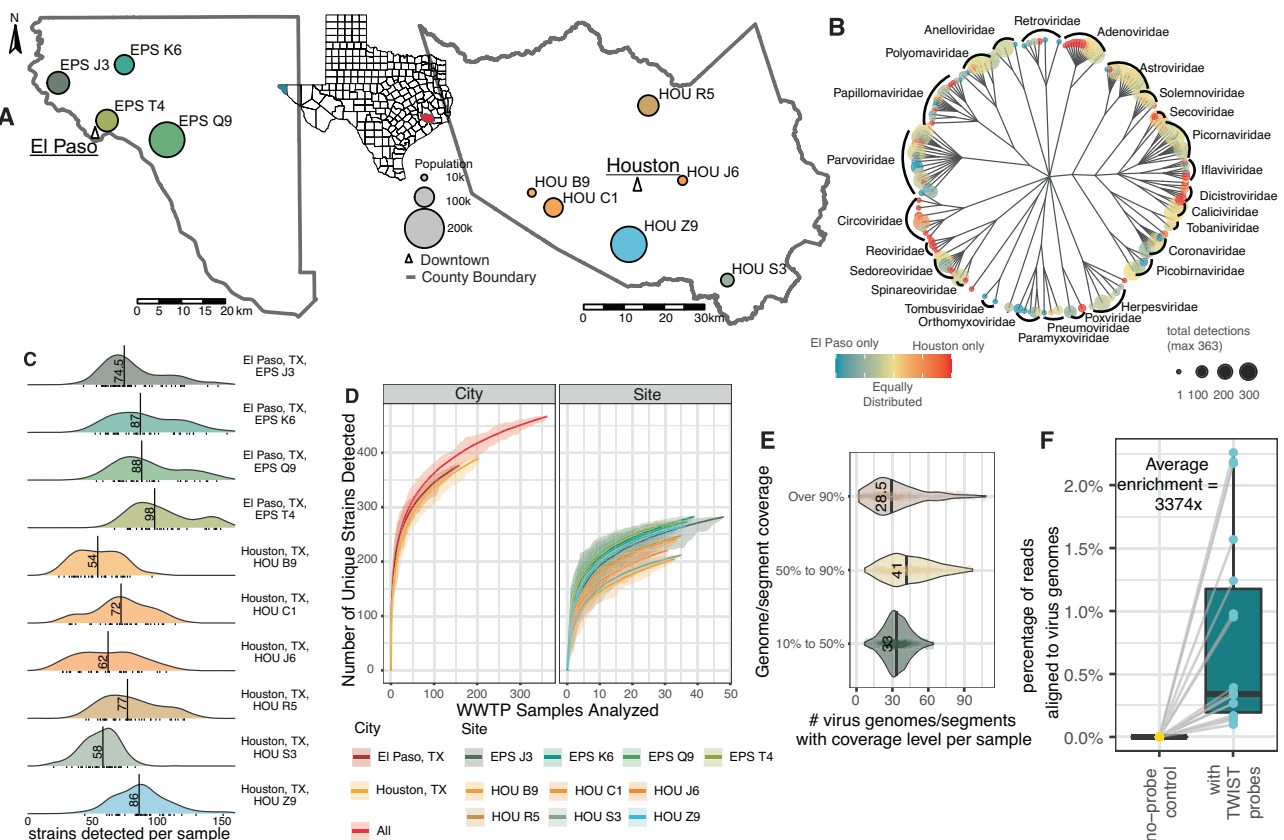

**Fig. 1 | Study sites, capture, and viral diversity. A** Map of wastewater catchment areas in Houston and El Paso, TX. The colored areas refer to the sites in each city (EP = 4, Houston = 6). **B** The treelike object was drawn with hierarchical taxonomical labels (kingdom, phylum, class, order, family, genus, species) rather than multiple sequence alignments due to independent origins of different virus phyla. Tip point size corresponds to number of wastewater samples with the virus detected, and color corresponds to the skew of the species to Houston (red) or El Paso (blue). **C** Number of distinct virus strains detected per sample from each wastewater treatment plant. **D** Rarefaction curves measuring distinct virus strains detected as more samples were analyzed. Lines represent average strains detected while shaded bands represent minimum and maximum values from 50 permutations. **E** Genome coverage of detected virus genome/segments for each sample. **F** Percentage of reads aligned to virus pathogen genome database in paired control (no-probe) and treatment (capture with the TWIST Comprehensive Virus Research Panel) groups. n = 18 biologically independent samples. Boxplots are defined as: center line = median, lower and upper box-bounds = 25th and 75th data percentiles, and whiskers extend to the minimum and maximum values.

and enterovirus D68. Interestingly, at times, there were different trends in virus levels observed in both cities and at different periods of the year (Fig. S4).

### Pathogenic virome communities follow spatiotemporal trends

We wished to understand how the human wastewater virome changed over space and time. Important variables in the structure of virome communities were realized by generating t-distributed stochastic neighbor embedding (t-SNE) plots from the virus abundance data of each sample. There was a stark separation of the samples by the city of collection and date of collection (Fig. 3A–B). Virus species from several families showed an uneven distribution between Houston and El Paso (Fig. S5A). For example, while we expect most viruses to have a prevalence bias towards El Paso due to higher median levels of strain detection per site (Fig. 1C), El Paso had especially strong signals from many Parvoviridae and Sedoreoviridae whereas Houston samples had higher prevalence of many Calicivirdae and Astroviridae, the reasons for which are currently unknown (Fig. S5A).

To assess community dynamics over time, all samples from each site were compared to each other using the Bray-Curtis dissimilarity statistic (Fig. 3C). In general, as time went on, the composition of the virome in samples diverged such that samples taken closer in time were quite similar, whereas those separated by many months were very different. Interestingly, a possible exception to the temporal divergence rule can be seen in samples taken from the wastewater

treatment plant serving Houston's large intercontinental airport, which likely reflects a transient population of world travelers (Fig. 3C, HOU R5). Here, the compositional dissimilarity was poorly correlated with the passing of time, possibly due to flux of the virome from incoming people. On the other hand, as the data collection approaches 1 year and the seasons repeat, samples from 3 of 4 El Paso sites seem to be re-converging on their community structures from the previous year. In general, dissimilarity follows a pattern where sites from different cities are more different than sites within the same city, and samples from the same site are more similar than everything else (Fig. 3D, Fig. S5B). Finally, we assessed the impact of human population size on virome diversity. The alpha diversity (Shannon's statistic) was measured for each sample (Fig. S5C), and the average diversity and population of the service area for each site were plotted (Fig. S5D). Average diversity increases from catchment populations of 10,000 to 100,000 inhabitants, but the diversity values level off with greater numbers of people. Collectively, this data confirms that the structure of wastewater virome communities are substantially determined by temporal and geospatial factors.

### Variant trends amongst the virome backdrop

A handful of viruses had high or complete genome coverage across many wastewater samples and were therefore suitable for variant analysis. Although a single lineage seemed to dominate the sample read abundance for some virus strains, many samples had a mix of two

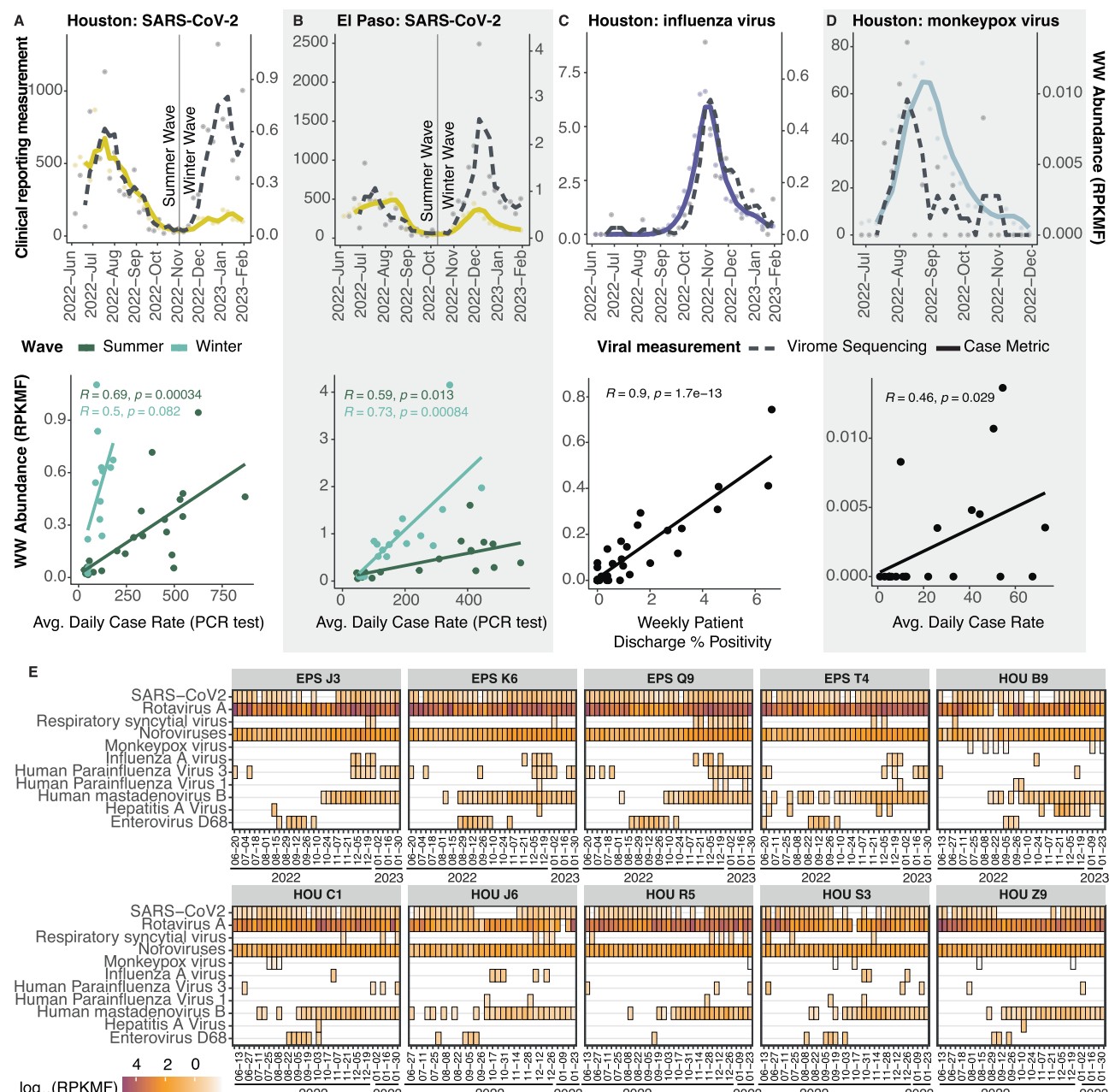

**Fig. 2 | Human viruses in wastewater correlate with clinical data. A** SARS-CoV-2 wastewater sequencing abundance compared to reported cases (top) and scatter plot with Pearson correlation coefficients and *p*-value for two-sided test between wastewater sequencing abundance compared to reported cases of SARS-CoV-2 (bottom) in Houston, TX. **B** SARS-CoV-2 wastewater sequencing abundance compared to reported cases (top) and scatter plot with Pearson correlation coefficients and *p*-value for two-sided test between wastewater sequencing abundance compared to reported cases of SARS-CoV-2 (bottom) in El Paso, TX. **C** Influenza wastewater sequencing abundance compared to reported Weekly Percentage of Visits with Discharge Diagnosed Influenza (top) and scatter plot with Pearson correlation coefficients and *p*-value for two-sided test between wastewater sequencing abundance compared to Weekly Percentage of Visits with Discharge Diagnosed of Influenza (bottom) in Houston, TX. **D** Monkeypox virus wastewater sequencing abundance compared to reported Mpox cases (top) and scatter plot with Pearson correlation coefficients and *p*-value for two-sided test between wastewater sequencing abundance. **E** Heatmap for all ten wastewater sites for presence/absence and abundance for 11 pathogens of major concern (*y*-axis) across the entire study period (*x*-axis).

or more lineages. Therefore, allelic variants were measured by the frequency of non-synonymous mutations compared to the reference genome. We focused on three examples.

Astrovirus MLB1, which has a seroprevalence in Americans close to 100%[26], was the virus contained at high genome coverage in the most samples in our dataset. The variant landscape of Astrovirus MLB1 was largely dictated by the city-of-origin of the sample (Fig. 4E), with gene-specific mutations in the capsid, ORF1a, and ORF1b showing strong regional localization in time (Fig. 4B, Fig. S6B). Human

Adenovirus 41 is an enteric virus associated with diarrhea and, possibly, hepatitis[27] in children and was also quite common in wastewater. This virus splits into two major lineages (Fig. 4A, D, Fig. S6A), with the hypervariable capsid (hexon) gene having a lot of diversity[28]. Although both lineages dominated in samples from either city, each city had one lineage that was more common. JC Polyomavirus, which is secreted in the urine, commonly establishes long, asymptomatic infections in a high proportion of the population[29]. Consistent with non-acute, rarely transmitted infections, the variant landscape of this virus seems to lack

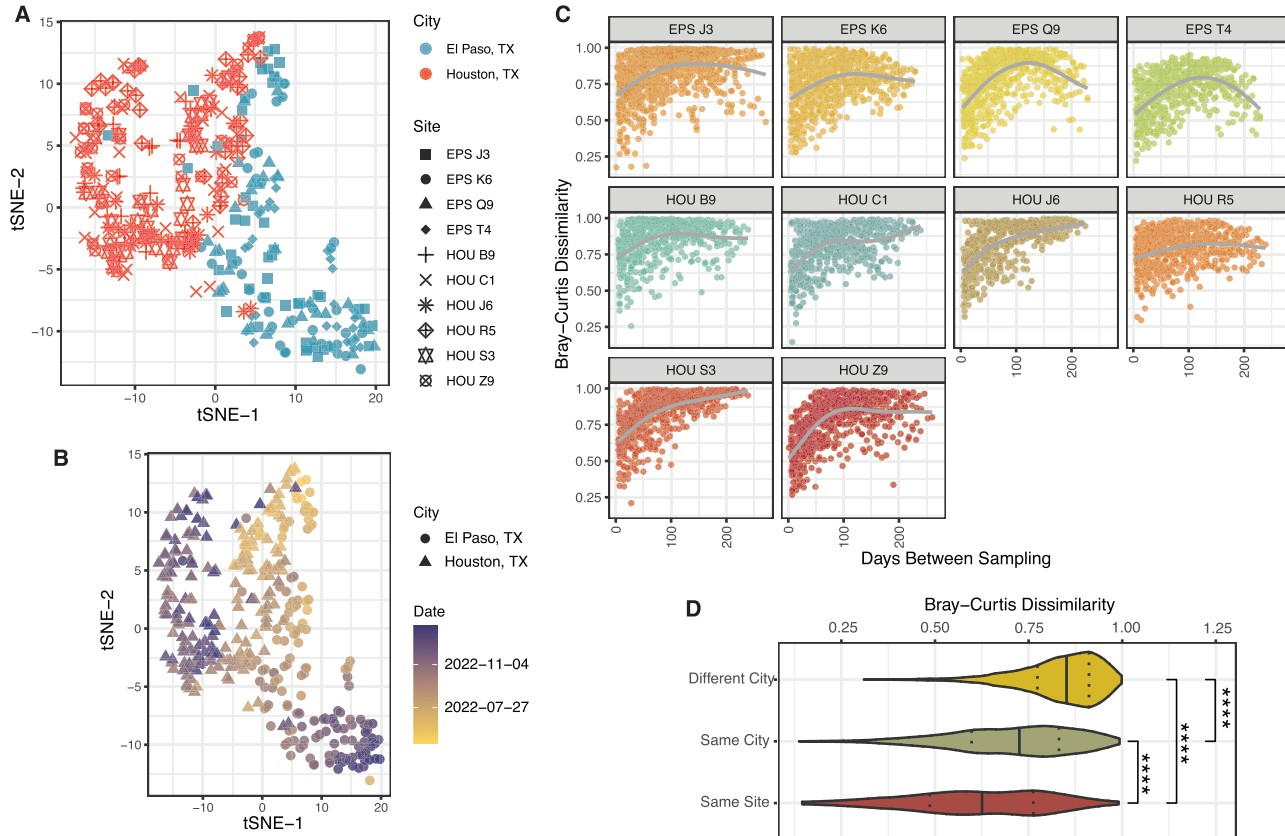

**Fig. 3 | Wastewater virome community structure. A** t-SNE of wastewater samples using virome abundance data, showing different cities/sites. **B** t-SNE of wastewater samples using virome abundance data, showing samples over time. **C** Temporal analysis of intra-site community changes. Each dot is a comparison between two samples. The *x*-axis measures days in between sampling. The *y*-axis measures Bray-Curtis dissimilarity between the samples. **D** Bray-Curtis dissimilarity between samples taken +/- 7 days apart, comparing samples from the same site, different site but same city, and different city. ****Represents *t*-test *p*-value < 1e-04. Different City vs Same Site, $p = 6.2e^{-164}$. Different City vs Same City, $p = 2.1e^{-214}$. Same City vs Same Site, $p = 2.9e^{-30}$.

meaningful spatiotemporal structure and most samples appear to have a diversity of lineages (Fig. 4C, F, Fig. S6C).

## Discussion

We present here an advance in our understanding of the human virome in wastewater. Undeniably, the detection of human viruses plays a crucial role in epidemiological surveillance, especially in a context where the allocation of resources for the testing and sequencing of clinical samples is dwindling. High level findings in this study include (i) the ability of a hybrid-capture probe approach that is designed against over 3,000 human and animal viruses to, when combined with deep sequencing, significantly enrich viral detection (465 total viruses detected in 6 months); (ii) the approach's suitability as a record of aggregate community infection levels for several key viruses of concern, including SARS-CoV-2, influenza virus, enterovirus D68, noroviruses, rotaviruses, monkeypox virus, respiratory syncytial virus and many others; (iii) the clear correlation of genome-specific sequencing reads to clinical data for SARS-CoV-2, influenza virus, and monkeypox virus; (iv) dynamics of the virome that change across space and time; and finally (v) the variant tracking of viruses with high genome coverage and prevalence.

Although this approach already offers a suitable estimation of relative virus levels, genomes of important pathogens such as SARS-CoV-2, influenza virus, and monkeypox virus were typically not sequenced at high enough coverage to assess allele and variant frequency. This was probably because these viruses do not typically carry high viral loads in the gastrointestinal or urinary tract and are not shed into wastewater in high numbers. Therefore, continued development

of the enrichment methodology as well as increased sequencing depth should be considered to improve genome coverage.

This comprehensive hybrid-capture methodology also holds promise for near real-time as well as retrospective assessments as it promises to detect virus genomes up to 15% different from known references. Also, additional probes can be logically designed and added to the existing reagent to capture highly divergent pathogens. In this way, we believe that, amongst other uses, this type of wastewater monitoring promises to be a useful, cheap, and scalable "smoke alarm" for populations under threat from dangerous pathogens.

Figure 2 seems to show the divergence in signal for SARS-CoV-2 between wastewater (both comprehensive sequencing and qPCR) and clinical PCR testing over time. This is almost certainly related to the decline in both symptomatic and asymptomatic PCR testing for SARS-CoV-2 and the rise of in-home rapid antigen testing coupled with overall decrease in testing in the United States.

The literature suggests that, while some SARS-CoV-2 variants were largely limited to certain geographies, major lineages (e.g., delta, omicron, BA.4) swept across the globe and were primarily separated by time[9], rather than geography. It is not clear whether this pattern observed for highly transmissible SARS-CoV-2 applies to other, less transmissible human viruses. Indeed, the data presented here for three highly abundant viruses suggests that this may not be the case (i.e., that geography is important).

The geographic partitioning between, for instance, allelic variants of astrovirus MLB1 but not JC polyomavirus (Fig. 4) may be due to transmission and infection modalities. Astroviruses transmit via a fecal-oral mechanism, and therefore are likely to propagate by local

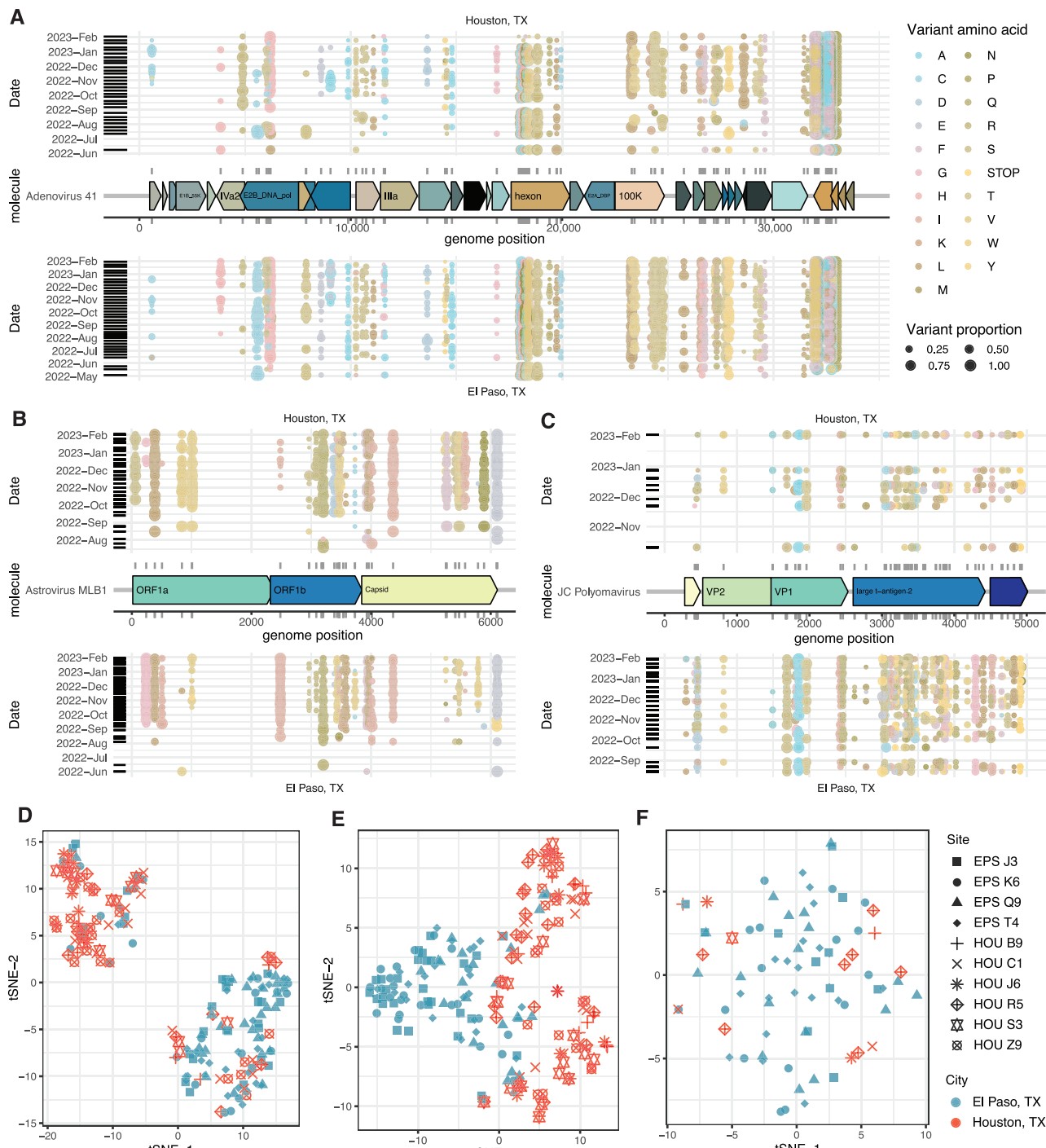

**Fig. 4 | Evaluation of non-synonymous variants in prevalent wastewater viruses. A** Genome map of Human Adenovirus 41 (middle) with non-synonymous variants displayed above (Houston, TX) and below (El Paso, TX) according to genome position (*X*-axis) and date (*Y*-axis). **B** Like (**A**) but with Astrovirus MLB1. **C** Like (**A**) but with JC Polyomavirus. **D** t-SNE of non-synonymous variant frequency of astrovirus MLB1. **E** Like (**D**) but with JC Polyomavirus. **F** Like (**D**) but with JC Polyomavirus.

outbreaks[30]. JC Polyomavirus is known to establish long asymptomatic infections in the large majority of people, probably being transmitted within households sporadically[29]. Astrovirus variants may be introduced to an immune-naive population or city and spread outbreak-by-outbreak over the course of months, whereas different JC polyomavirus lineages acquired by different people over many years could be shed simultaneously, blurring allelic patterns seen with other viruses. It is tantalizing to consider that such an approach may help us understand the underlying infection etiology of these and related viruses as it pertains to entire population transmission and dynamics.

The distinct spatial patterns of virus communities (Fig. 3) may be caused by several phenomena including weather, demographics, the structure of human habitation, vaccination rates, and previous exposure history. The temporal trends may be driven by similar factors along with emergence of fitter virus variants and cyclical/annual pathogen patterns.

The future of wastewater pathogen monitoring is promising, but responsible parties will need to refine methods to provide timely, comprehensive, and useful information[6]. This study shows that some of these attributes are already being attained, but additional innovation and interest will be required to get up to speed on all fronts. Post-pandemic society will require orthogonal, comprehensive viral surveillance of distinct pathogen reservoirs, including wastewater, to better equip epidemiologists with accurate information for public health action.

## Methods

Our research study complies with institutional review boards from University of Texas MD Anderson Cancer Center and Baylor College of Medicine.

### Sample collection and shipping

Between 100–500 mL of raw wastewater was collected into 500 ml leak-proof prelabeled sample bottles at 6 wastewater treatment plants in Houston, TX and 4 in El Paso, TX. Treatment plants were coded upon the request of public health officials. The surface of the sample bottles was decontaminated with 10% bleach and moved to a "clean" zone, where the samples were sealed into biohazard bags in shipping boxes with absorbent pads and ice packs for overnight shipping to the Alkek Center for Metagenomics and Microbiome Research at Baylor College of Medicine, Houston, TX.

### Sample processing and nucleic acid extraction

Wastewater samples were barcoded upon arrival and stored at 4 °C until processing. First, 50 mL of wastewater was decanted and centrifuged at 3374 x g for 10 min, separating the solid and liquid fractions. The supernatant was then vacuum filtered using an ion-based cellulose filter paper and the virus-containing cellulose filter was placed into a bead-beating tube with lysis buffer. The tube was run on the homogenizer for 1 min at 5 m/s, rested for 1 min, then run on the homogenizer for 1 more minute. Following the bead beating the samples were centrifuged at 14–17×1000 RPM for 2 min. DNA and RNA were extracted using the Qiagen QIAamp VIRAL RNA Mini Kit.

### Library preparation, probe-based virome capture and sequencing

RNA extracts were converted to cDNA using Protoscript II First Strand cDNA Synthesis Kit (New England Biolabs Inc.), NEBNext Ultra II Non-Directional RNA Second Strand Module (New England Biolabs Inc.), and Random Primer 6 (New England Biolabs Inc.). A total of 25 ng of the cDNA and DNA mix was used for library construction using Twist Library Preparation EF 2.0 Kit and Twist Universal Adaptor System (Twist Biosciences). The libraries were pooled, a maximum of 16 samples per pool, at equal mass to a total 1,500 ng per pool. The Twist Comprehensive Viral Research Panel (Twist Biosciences) was used to hybridize the probes at 70 °C for 16 h. The post-capture pool was further PCR amplified for 12 cycles and final libraries were sequenced on Illumina NovaSeq 6000 SP flow cell, to generate 2×150 bp paired-end reads. Following sequencing, raw data files in binary base call (BCL) format were converted into FASTQs and demultiplexed based on the dual-index barcodes using the Illumina 'bcl2fastq' software.

### Reverse transcription Quantitative PCR (RT-qPCR) of Influenza A and SARS-CoV-2

RT-qPCR for Influenza A and SARS-CoV-2 was carried out using CDC FluSC2 assay (https://www.cdc.gov/coronavirus/2019-ncov/lab/multiplex.html) with a total 20 μl volume containing 5 μl of eluted RNA and 15 μl of TaqPath 1-step Multiplex Master Mix, (A28523 Applied Biosystems) under the following cycling conditions: 25 °C for 2 min, 50 °C for 15 min, 95 °C for 2 min, and 45 cycles of 95 °C for 3 s, then 55 °C for 30 s on a 7500 Fast Dx Real-Time PCR Instrument (4406985,

Applied Biosystems) with SDS version 1.4 software. The fluorescent signal was measured during the annealing step. Primer sequences and concentration of primers and probes used are available in Supplemental Table 1. Samples were run in duplicate wells and included negative extraction control and no template control. Samples were considered positive if Ct values were <45. To determine copy numbers, oligonucleotide standard targeting Influenza A matrix protein was designed and purchased from IDT technologies. For SARS-CoV-2, a standard curve of the linearized N plasmid (SARS-CoV-2 (2019-nCoV) RUO Plasmid Controls) was purchased from IDT technologies and used to determine the genomic copy numbers. The Oligonucleotide standard curve for Influenza A ranging from $71 \times 10^6$–710 copies/mL and 16,000-16 copies/mL were run and average CT values of the duplicate wells were used to determine the copy numbers using the standard curve.

### Preparation of RSV stocks and determining copy numbers of RSV stock

The RSV-A used in the spike-in experiments was isolated from an adult with RSV infection in 2015. Demographics and clinical information were prospectively collected from structured interviews and review of medical records of the subject. The institutional review board of the University of Texas MD Anderson Cancer Center and Baylor College of Medicine approved the study and informed consent was obtained from the participant to test and isolate the sample and consent for future use. Nasal wash sample was inoculated on HEp-2 cells (CCL-23, ATCC). HEp-2 cells were maintained in Eagle's Minimum Essential Medium (MEM), (Mediatech) supplemented with 10% fetal bovine serum (FBS) (Hyclone), 2 mM glutamine and antibiotic and antimycotic (Gibco Life Technologies). After initial isolation, the working stock was prepared by passing RSV four times to make working pools. Viral working stocks are made by infecting Hep-2 cells at a multiplicity of infection of 0.1 in MEM supplemented with 2% FBS, 2 mM glutamine and antibiotic and anti-mycotic. Infection is monitored for 2–3 days until 75% cytopathic effect is achieved. RSV is harvested by disrupting the cell monolayer with sterile glass beads and the resulting cell suspension was sonicated at 60 W for 2 min (Branson sonicator). The viral suspension was clarified by centrifugation and diluted 1:1 in Iscoves modified Dulbecco's medium and 15% glycerol. Viral aliquot stocks of 1 ml were snap frozen in ethanol and dry ice bath, and stored at −70 °C. The viral stocks contained $1.53 \times 10^6$ plaque-forming unit/ml. Copy number for the viral stocks were determined using RT-qPCR. Viral RNA was extracted from RSV stock and real-time qPCR was performed in duplicate wells. RT-qPCR for RSV-A was performed using Ag-Path One step-RT-PCR Master Mix (4387391 Applied Biosystems) under the following cycling conditions: 45 °C for 10 min, 95 °C for 10 min, and 45 cycles of 95 °C for 15 s, then 55 °C for 1 min. Primer sequences and concentration of primers and probes used are available in Supplemental Table 1. Oligonucleotide standards targeting RSV nucleoprotein (N) sequence were designed and were purchased from IDT technologies. Oligonucleotide standard curve for RSV N ranging from $85 \times 10^6$ - 850 copies/mL were run and average CT values of the duplicate wells were used to determine the copy numbers using the standard curve.

### Geographic shape files

Shape files for Texas counties and cities were downloaded from the US Census TIGER program repository (https://www.census.gov/geographies/mapping-files/time-series/geo/tiger-line-file.2021.html#list-tab-790442341).

### Virus pathogen database compilation

The TWIST Comprehensive Virus Research Panel is reported to contain over 1 M unique probes that enable detection of 3,153 viral human and nonhuman pathogens. To construct the Virus Pathogen Database

(v2.0.2), all available complete isolate genomes for each viral genus covered by the TWIST Comprehensive Virus Research Panel were downloaded from NCBI's GenBank using Datasets (https://www.ncbi.nlm.nih.gov/datasets/) on November 16th, 2022. Ninety-seven virus genomes were determined to be of highest public health concern based on the subjective opinion of the authors. These genomes were protected from de-replication and segments were concatenated when relevant (available at https://zenodo.org/record/7876309). All other virus genomes and segments were de-replicated at 95% average nucleotide identity and 85% alignment fraction using two rounds of the anicalc/aniclust scripts from CheckV[31]. Exemplar sequences for each cluster were kept in the database (https://zenodo.org/record/7876309). Hierarchical taxonomic data was obtained for each sequence from the kingdom level to the strain level using Taxonkit[32].

### Sequencing read processing for virus genome and segment detection

Demultiplexed raw fastq sequences were processed using BBDuk to quality trim (Q25), remove Illumina adapters, and filter PhiX reads. Reads with a minimum average Phred quality score < 23 and length < 50 bp after trimming were discarded. Trimmed FASTQs were mapped to a combined PhiX (standard Illumina spike in) and human reference genome database (GCF_000001405.39) using BBMap to determine and remove human/PhiX reads[33]. Processed reads were run through the EsViriru pipeline (v0.1.1) with default settings. Specifically, reads were aligned to the entire Virus Pathogen Database (v2.0.2) via minimap2[34,35] and alignments were filtered by CoverM (https://github.com/wwood/CoverM) to require at least 90% average identity across 90% of the read length. Virus genomes/segments with reads covering at least either 1000 nucleotides or 50% of the genome/segment length were considered preliminary detections. Even de-replicated virus sequence sets can have regions that are highly similar between two or more sequences. Consensus sequences from this preliminary set of detected virus genomes/segments were extracted using samtools consensus[35]. After removing strings of ambiguous N's, the preliminary consensus sequences were compared to each other pairwise using anicalc, and clusters were made with sequences of at least 98% average identity using aniclust. The longest consensus sequence in each cluster was kept as the final sequence. The fastq reads were then aligned to a smaller database of only the references corresponding to final sequences with the same parameters for minimap2/CoverM. Virus genomes/segments from this alignment with reads covering at least either 1000 nucleotides or 50% of the genome length were considered final detections. The metric RPKMF was calculated as (Reads Per Kilobase of reference genome)/(Million reads passing Filtering). The more common RPKM was not used here and is calculated as (Reads Per Kilobase of reference genome)/(Million reads mapped to reference sequences). This metric was used because, at low levels seen in wastewater, the proportion of reads aligning to virus genomes is presumed to correspond to the proportion of viral nucleic acid molecules over background molecules in the physical sample. Finally, per sample abundance and coverage metrics and taxonomic information were gathered in tabular format and merged into a single table for downstream processing.

### Virus community and variant analyses

The tables reporting virus abundance in RPKMF and sample metadata were processed in R, using tidyverse packages. Alpha diversity and Bray-Curtis dissimilarity statistics were calculated using Vegan (https://github.com/vegandevs/vegan). Plots were made with ggplot[36], ggpubr (https://github.com/kassambara/ggpubr), tSNE (https://github.com/jkrijthe/Rtsne), and color packages wesanderson (https://github.com/karthik/wesanderson) and nationalparkcolors (https://github.com/katiejolly/nationalparkcolors). Reads aligning to analyzed reference genomes were processed using iVar[37] with default settings and

filtered to only report non-synonymous mutations with an allele frequency of at least 3% and a minimum allele coverage of five reads. Dendrogram figures were drawn in R with ggtree[38], aplot and ggplot.

### Wastewater virome sequencing correlation with RT-PCR and clinical data

For each virus analyzed, the wastewater virome abundance was summarized as the mean RPKMF of the relevant genome for all wastewater treatment plants in each city for each week. For Influenza Virus, average RPKMF values for H3N2 and H1N1 were added together. For RT-PCR, genome copies (described above) were averaged across all wastewater treatment plants in each city for each week. SARS-CoV-2 clinical PCR-positive case data and test positivity data were averaged across each week. The data for Houston were obtained at (https://covid-harriscounty.hub.arcgis.com/). The data from El Paso were obtained from El Paso Public Health upon request. Influenza "Weekly Percentage of Visits with Discharge Diagnosed Influenza" values were manually recorded from weekly Houston Health Department Flu Reports (https://www.houstonhealth.org/services/data-reporting/flu-reports). Monkeypox case numbers were scraped from the Texas Department of State Health Services website and transformed into weekly average cases. After transformation to weekly data, Simple moving average plots with an averaging period of 3 weeks were used to visualize temporal trends with R package tidyquant (https://github.com/business-science/tidyquant). Pearson statistics were used to quantify correlations between data.

### Reporting summary

Further information on research design is available in the Nature Portfolio Reporting Summary linked to this article.

## Data availability

The data used in this study's analyses is deposited at Zenodo repository https://zenodo.org/record/7884454, https://doi.org/10.5281/zenodo.7884454 (follow instructions on the GitHub repository). All sequencing reads are uploaded to SRA at accession PRJNA966185 with any human sequences removed.

## Code availability

All analyses for this manuscript can be reproduced with the code at https://github.com/cmmr/TX_wastewater_virome with data from the Zenodo repository referenced in the Data Availability section. EsViritu is available at https://github.com/cmmr/EsViritu (see this GitHub repository for most recent databases).

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

## Acknowledgements

The authors thank the Health Departments and Water Utilities of Houston and El Paso for their support in the contribution of wastewater samples to the project. We also acknowledge Adrien Assie for illustrating the logo for EsViritu. This work was supported by S.B. 1780, 87th Legislature, 2021 Reg. Sess. (Texas 2021) (E.B., A.W.M., and J.F.P.), NIH/NIAD (Grant number U19 AI44297) (A.W.M.), and Baylor College of Medicine and Alkek Foundation Seed (J.F.P.).

## Author contributions

Conceptualization: M.T., S.J.C., J.R.C., T.A., J.C., F.W., J.R., J.D., B.H., C.X.B., K.D.M., P.A.P., A.W.M., C.T., K.L.H., J.F.P., E.B.. Methodology: M.T., S.J.C., V.A., P.Z., T.A., K.F., K.L.H., M.C.R., J.B., P.A.P., D.H., A.G/, K.D.M., H.M., K.P., L.W.. Investigation: M.T., S.J.C., V.A., A.W.M.. Visualization: M.T., K.Z., R.L., C.X.B.. Funding acquisition: J.F.P., E.B., A.W.M.. Project administration: S.J.C., M.C.R., J.C., J.B., K.D.M., A.W.M.. Supervision: S.J.C., J.F.P., E.B., A.W.M.. Writing—original draft: M.T., S.J.C., A.W.M.. Writing—review and editing: M.T., S.J.C., J.C., F.W., J.R., J.D., C.X.B., P.A.P., J.F.P., E.B., A.W.M., C.T..

## Competing interests

The authors declare that they have no competing interests.

## Additional information

[1]The Alkek Center for Metagenomics and Microbiome Research, Department of Molecular Virology and Microbiology, Baylor College of Medicine, Houston, TX 77030, USA. [2]Department of Molecular Virology and Microbiology, Baylor College of Medicine, Houston, TX 77030, USA. [3]TAILOR Labs, Baylor College of Medicine, Houston, TX 77030, USA. [4]School of Public Health, University of Texas Health Science Center at Houston, Houston, TX 77030, USA. [5]Texas Epidemiologic Public Health Institute (TEPHI), Houston, TX, USA. [6]Department of Epidemiology, Human Genetics and Environmental Sciences, UTHealth Houston School of Public Health, Houston 77030, USA. [7]El Paso Water Utility, El Paso, TX, USA. [8]Department of Biostatistics and Data Science, UTHealth Houston School of Public Health, Houston, TX 77030, USA. [9]Center for Spatial-temporal Modeling for Applications in Population Sciences, UTHealth Houston School of Public Health, Houston, TX 77030, USA. [10]Department of Pediatrics, Baylor College of Medicine, Houston, TX 77030, USA. [11]These authors contributed equally: Michael Tisza, Sara Javornik Cregeen. ✉e-mail: jpetrosi@bcm.edu; Eric.Boerwinkle@uth.tmc.edu; maresso@bcm.edu

