## [Peer Review File · Nature Communications]

REVIEWER COMMENTS

Reviewer #1 (Remarks to the Author):

In this manuscript, the authors carry out a sequencing-based analysis of 363 longitudinal wastewater samples collected from ten diverse sites within two significant urban areas. This investigation into the human virome leverages a viral probe capture set and develops a bioinformatics pipeline purposed for virus detection. The researchers draw correlations between viral sequencing data and clinical cases, revealing spatiotemporal trends within viral communities. Furthermore, they demonstrate regional spread and evolutionary patterns in the three most prevalent viruses. Undeniably, the detection of human viruses plays a crucial role in epidemiological surveillance, especially in a context where the allocation of resources for the testing and sequencing of clinical samples is dwindling. The usage of wastewater-based epidemiology for virus monitoring could be pivotal for the prevention and control of diseases. However, I have some major concerns that need to be addressed and clarified thoroughly to enhance the quality and significance of this study.

Major comments:

- Regarding the title, the term 'comprehensive' might not accurately portray the content of this manuscript. Although the authors claim to have detected multiple viruses using sequencing methods, they have not sufficiently elaborated on this aspect within the body of the paper, instead focusing predominantly on a select few. Therefore, the usage of 'comprehensive' creates ambiguity and may not represent the manuscript's actual scope accurately. It is recommended that the authors revisit this terminology, ensuring it aligns more precisely with the presented data and discussions.

- The authors have not reported the sensitivity and specificity of the commercially available probe-based capture method employed in the study. Although a comparison has been made between the viral reads counts from this method and those from unenriched samples, demonstrating significant sequence enrichment (claimed to be 3374-fold), I believe that the authors should provide an evaluation of the actual performance of this method, especially with respect to viruses of major interest. The method may be developed based on clinical samples, but wastewater samples exhibit greater matrix effects, which can significantly complicate hybridization. Moreover, there might be concerns about cross-reactions from multiple probes, leading to false positive and negative results. This could potentially be demonstrated by spiking standard controls of specific viruses of concern into wastewater samples deemed negative, and then comparing their recovery. The inclusion of such data would be of high value and practical use to future users of this method.

- Line 111-112

The details and profiles of the '465 distinct virus strain' should be provided in the main text or supporting information. Besides, I am rather confused about the frequency and overlap of these detected viruses at different sites. Is it possible to relate and connect this with the 'population health' mentioned in the abstract? Could the authors elaborate on the deeper implications of these findings in the context of public health?

- Line 167-168

Is there a connection to the performance of the method used in the study? Could the sensitivity of the method be provided?

- Line 169-173

What patterns or findings can be seen across different sites and periods? How are the observed trends at the virus level explained or correlated, and what further significance or implications might have?

- Line 216-218

The possible reasons for the findings should be discussed.

- Line 234-236

Does it seem the discussion not align well with the situation of Houston?

- Line 298

For the (v) finding, is the low completeness of genomes related to the sensitivity of the method?

- Line 302-305

The enrichment effect of the probe-based capture method on viruses of high concern, such as SARS-CoV-2 or influenza virus, should be demonstrated in the manuscript.

Minor comments:

- Fig.2

The caption for Figure 2E appears to be missing.

- Figure 4A-4C

The presentation of information in these figures is not particularly clear and could benefit from further refinement.

- Line 341

The centrifugation parameters should be elucidated.

- Regarding the real-time PCR, the manuscript should comply with MIQE guidelines (Minimum Information for Publication of Quantitative Real-Time PCR Experiments; <https://doi.org/10.1373/clinchem.2008.112797>).

- Line 380

The term 'subjective opinion' is unclear in this context.

- Line 390

Why is the 'quality score < 23' selected?

- Line 398

Why do you use 'either 1000 nucleotides or 50% of the genome/segment length'?

- Line 432

Is 'RT-PCR' used in this context, or is it RT-qPCR/Real-time RT-PCR?

- Line 434

The verb 'was averaged' should be 'were averaged' to agree with the plural subject.

- Table 1

The direction of the sequences (e.g., 5'-3') should be indicated in this table.

Reviewer #2 (Remarks to the Author):

Wastewater-based epidemiology (WBE) is an upcoming field, which contributed significantly to the understanding of virus transmission during the pandemic. Several studies have used viral titers in wastewater to predict the trajectory of the pandemic. WBE is thus an important tool, which needs to be refined for future applications in public health.

This study is a good example of applied WBE. The authors have refined the technique by selectively enriching clinically relevant human viruses. The technique allowed a deeper insight into the diversity of human viruses in wastewater.

The spatiotemporal distribution of viruses is clearly evident from the results. In the future, it will be interesting to elucidate the factors responsible for this distribution.

Overall the study is an important contribution to this field, which would improve our understanding of viral transmission and its effect on human health.

Minor revisions: 1. In Fig.2, the legend for panel E is missing.

2. In the discussion, a section explaining why the numbers are underestimated using qPCR compared to virome sequencing may be included. This is most likely because PCR positivity represents only symptomatic cases. Also not every symptomatic case is tested for qPCR positivity. On the other hand, WBE enables the detection of viruses shed by a large population of asymptomatic cases.

We thank the reviewers for their constructive criticisms and suggestions. We believe the manuscript is much improved by their reviews. Based on their remarks, we have taken experimental and analytical steps to improve the manuscript. Our improvements include defining the dynamic range of our quantitative wastewater virome detection methodology, expounding upon claims and analyses regarding the comprehensiveness of our approach, and discussing the broader implications of our findings. We respond point-by-point below, with the reviewers' comments highlighted in yellow, and our answers in plain text.

Note: RTR = Response to Reviewer.

REVIEWER COMMENTS

Reviewer #1 (Remarks to the Author):

In this manuscript, the authors carry out a sequencing-based analysis of 363 longitudinal wastewater samples collected from ten diverse sites within two significant urban areas. This investigation into the human virome leverages a viral probe capture set and develops a bioinformatics pipeline purposed for virus detection. The researchers draw correlations between viral sequencing data and clinical cases, revealing spatiotemporal trends within viral communities. Furthermore, they demonstrate regional spread and evolutionary patterns in the three most prevalent viruses. Undeniably, the detection of human viruses plays a crucial role in epidemiological surveillance, especially in a context where the allocation of resources for the testing and sequencing of clinical samples is dwindling. The usage of wastewater-based epidemiology for virus monitoring could be pivotal for the prevention and control of diseases. However, I have some major concerns that need to be addressed and clarified thoroughly to enhance the quality and significance of this study.

Major comments:

- Regarding the title, the term 'comprehensive' might not accurately portray the content of this manuscript. Although the authors claim to have detected multiple viruses using sequencing methods, they have not sufficiently elaborated on this aspect within the body of the paper, instead focusing predominantly on a select few. Therefore, the usage of 'comprehensive' creates ambiguity and may not represent the manuscript's actual scope accurately. It is recommended that the authors revisit this terminology, ensuring it aligns more precisely with the presented data and discussions.

RESPONSE: We appreciate Reviewer #1's careful reading and evaluation of the manuscript. As constructed, and to the reviewer's point, we may not have adequately conveyed the comprehensiveness afforded by the methodology. Therefore, to rectify this, we have added new text and a new panel to Fig. 1 (also shown here as Fig RTR1) and commented on this result in lines 41 and 112-113 of the revised manuscript.

First, we hold committed to the notion this effort is comprehensive. In all, distinct virus genomes from 28 families, 77 genera, 191 species, and 465 strains were detected in one or more wastewater samples. Simply put, we know of no other study reporting this level of breadth of viral detection and analysis in wastewater. To illustrate this point further, we now provide a tree-like relationship (Fig RTR1) that demonstrates the breadth, frequency, and distribution between Houston and El Paso of these families.

Second, both a spatial (two major cities) and longitudinal (every week for nearly a year) analysis was performed. These two attributes add further comprehensiveness to the analysis since many of the patterns and trends were consistent between cities.

Third, the analysis is not only scientifically rigorous (continuous longitudinal analysis as described above and reproducible), but a second orthogonal analysis (RT-qPCR) was conducted at the same time for 12 of the main viruses of concern. The correlation of the RT-qPCR signal in several cases with the sequencing signal adds a comprehensiveness in rigor not typically observed in such reports. Taken together, we believe that these analyses demonstrate the overall comprehensive capability of our methodology and justify the use of word 'Comprehensive' in the manuscript title.

Figure RTR1

Taxonomical representation of detected viruses. The treelike object was drawn with taxonomical labels (kingdom, phylum, class, order, family, genus, species) rather than multiple sequence alignments due to independent origins of different virus phyla. Tip point size corresponds to number of wastewater samples with the virus detected, and color corresponds to the skew of the species to Houston (red) or El Paso (blue).

- The authors have not reported the sensitivity and specificity of the commercially available probe-based

capture method employed in the study. Although a comparison has been made between the viral reads counts from this method and those from unenriched samples, demonstrating significant sequence enrichment (claimed to be 3374-fold), I believe that the authors should provide an evaluation of the actual performance of this method, especially with respect to viruses of major interest. The method may be developed based on clinical samples, but wastewater samples exhibit greater matrix effects, which can significantly complicate hybridization. Moreover, there might be concerns about cross-reactions from multiple probes, leading to false positive and negative results. This could potentially be demonstrated by spiking standard controls of specific viruses of concern into wastewater samples deemed negative, and then comparing their recovery. The inclusion of such data would be of high value and practical use to future users of this method.

In principle we agree with the reviewer and have conducted additional experimentation to address this point. To address the comment about sensitivity of this hybrid-capture-based assay for viruses of concern in wastewater, we conducted a serial dilution experiment with Respiratory Syncytial Virus A (RSV-A). RSV-A was chosen because it is of great concern to many public health experts due to its effect on children and the elderly, was recently reported as being found in wastewater (making its monitoring in this medium of potential high value), and because our team has access to highly purified and quantified stocks of the virus. We used wastewater samples that were previously sequenced with our methods and chosen based on absence of RSV-A. Lab stocks of RSV-A were quantified to establish concentration of genome copies. Then, eight 5:1 serial dilutions were spiked into unprocessed wastewater samples, with estimated genome copies ranging from 4 million copies to 51 copies. These samples were then run through our regular nucleic acid extraction, hybrid-capture, and sequencing pipeline described in this manuscript.

The data, as represented in Figure RTR2, demonstrates that our methodology is exquisitely quantitative for RSV-A genomes. In fact, the method is so sensitive that we detected as few as 51 genome copies in a 50ml wastewater sample (Fig. RTR2), and the input genome copy number is highly correlated with the output abundance value (RPKMF) from the sequencing library ($R = 0.998$ with untransformed data and $R=0.975$ with log transformed data) (Fig. RTR2). These results are similar to TWIST's technical document using the same probes thereby providing strong evidence that this approach is very sensitive (<https://www.twistbioscience.com/products/ngs/fixed-panels/comprehensive-viral-research-panel?tab=resources>). Since multiple probes are present for many of the viruses on our list, we believe the sensitivity may be enhanced by having non-redundant capture probes specific for the same strain or species of virus, which should now be regarded as one benefit of using sequencing-based detection from probe-captured nucleic acid. Future planned studies will use other viruses in similar dilution experiments to bolster these findings, work that is beyond this study because it takes time and resources to produce human virus from key viruses of concern. This work is now represented in the revised manuscript as Fig. S2 and we comment on this sensitivity and result in Lines 121-125.

Figure RTR2

RSV-A spike-in

Dynamic range of pathogen detection in wastewater using RSV spike-in. All charts represent the results of an experiment spiking RSV into real wastewater samples. Wastewater samples were first processed with the hybrid-capture method described in this paper to screen for presence of RSV and only RSV-negative samples were used here. A 5:1 dilution series, starting with 4 million estimated genome copies down to 51 estimated genome copies, was prepared with wastewater samples, then processed with hybrid capture methods, sequenced, and quantified with EsVirtu.

- Line 111-112

The details and profiles of the '465 distinct virus strain' should be provided in the main text or

supporting information. Besides, I am rather confused about the frequency and overlap of these detected viruses at different sites

Thank you. As you might imagine, when a project generates this much information, the challenge becomes how to show it in ways meaningful for readers. This is the reason we focused on the pathogenic viruses so the reader would not lose sight of the value of this work.

To address this issue here, we have taken a few steps. First, we believe we addressed the frequency of each virus species in each city in Figure RTR1 which is now Fig. 1B in the revised manuscript. This gives the readers a better view of the broad nature of the detection we are observing. Second, we thought it might be helpful to assess the correlation between all pairs of virus species. We found several correlational clusters (Fig. RTR3) that have not been detected or studied previously. This figure shows a heat map of their level as they relate to each other. In doing this, we see that these associations are vast and complicated and thus believe it's best to leave this out of the MS and only show it for the reviewers. We think more clear associations and trends might originate after multiple years of wastewater monitoring which might be more intriguing to discuss. Additionally, we hope and expect that this data may be reanalyzed by interested readers, and we've made the read data available at NCBI Bioproject PRJNA966185 and the abundance tables, with each detected virus in every sample, available as "wastewater_virome_abundance_table1.tsv" at <https://zenodo.org/record/7884454>.

Figure RTR3

Is it possible to relate and connect this with the ‘population health’ mentioned in the abstract? Could the authors elaborate on the deeper implications of these findings in the context of public health?

The public health implications are multifaceted. To begin with we enjoyed the language used in Reviewer #1’s summary statement: “Undeniably, the detection of human viruses plays a crucial role in epidemiological surveillance, especially in a context where the allocation of resources for the testing and sequencing of clinical samples is dwindling”. We believe we did correlate some of this data to public health. We were surprised at how well some of the sequencing signals for some of the pathogenic virus species correlated to the clinical data for those viruses. We suspect that if we had much more clinical

data for other viruses it would also correlate well with the wastewater sequencing signals. So, it is our belief that, at the moment, this is the most useful way to use this data for public health, namely, that the wastewater virome sequencing pioneered in this study offers faster, less biased, more comprehensive, and cheaper ways to assess the burden of specific viral diseases in cities and communities than traditional RT-qPCR-based wastewater assessment or traditional epidemiology. Furthermore, the approach will bolster clinical observations by being a separate orthogonal way to confirm those observations.

Finally, and of great use with this technology, will be our ability to detect the emergence of new dangerous viruses, their variants, or viruses of catastrophic concern (e.g. Ebola or Smallpox). Inevitably, there will be outbreaks of these or new viruses that will spread across communities or even the globe in the future. Only through a comprehensive sequencing-based approach like the one demonstrated here will we be best positioned to detect such a virus, and even in real-time provide some sequence information on it. The future of this work is exciting. For example, the approach can be readily modified with additional oligos designed against a new threat and built into the pipeline. We've added some of these points to the revised manuscript - lines 337-342.

- Line 167-168

Is there a connection to the performance of the method used in the study? Could the sensitivity of the method be provided?

We addressed this in the response above (Figure RTR2).

- Line 169-173

What patterns or findings can be seen across different sites and periods? How are the observed trends at the virus level explained or correlated, and what further significance or implications might have?

- Line 216-218

The possible reasons for the findings should be discussed.

A major goal of long-term wastewater-based epidemiology is to answer these very questions. While we cannot provide definitive answers at this time, we propose the following factors as possible explanations: weather, demographics, the structure of human habitation, vaccination rates and previous exposure history, and the complexity associated with human social interactions and travel. We have added text in the revised manuscript (lines 365-368) that address these possibilities.

One of the more fascinating results that may soon be observed is whether or not these trends show a universal "virome seasonality." Its too early to say conclusively, but we are now observing resemblance of the current signal to the signal at this time last year.

In addition, we refer the reviewer to the answer in response to the reviewer's comment about connections to public health (above).

- Line 234-236

Does it seem the discussion not align well with the situation of Houston?

This is an astute observation by the reviewer. We also find this pattern striking. We note that Houston has a range of sites serving ~10,000 residents to sites serving >300,000 residents whereas El Paso has

three sites serving ~100,000 residents and one site serving ~400,000 residents. It is possible that population effects on Shannon Diversity reach saturation at around 100,000 residents, then other, unknown factors become more important such as proximity to zoonotic animals or average age of the population. Assessing diversity metrics in more sites from more cities will address these speculations.

- Line 298

For the (v) finding, is the low completeness of genomes related to the sensitivity of the method?

Yes, we believe so. The more genome coverage in the signal, the greater the signal strength. This speaks to the importance of the comment made above about the probes. The more probes covering amplifiable regions of the genome, the more likely we will detect that virus and probably the greater the reproducibility of the signal strength amongst multiple samples or repeats of the same sample. Note that in theory it is also possible a really good coverage of a small portion of the genome (i.e. local strong annealing of probes) can also lead to good signals. All of these ideas will be tested as this science develops over the coming years.

- Line 302-305

The enrichment effect of the probe-based capture method on viruses of high concern, such as SARS-CoV-2 or influenza virus, should be demonstrated in the manuscript.

We address this in Figure RTR2. Also, please see the original Fig 1F that shows how probe enrichment is critical for the overall signal for the entire virome (No viruses of high concern, such as SARS-CoV-2 were detected in the “no-probe” samples).

Minor comments:

- Fig.2

The caption for Figure 2E appears to be missing.

We have now added the caption for Figure 2E.

- Figure 4A-4C

The presentation of information in these figures is not particularly clear and could benefit from further refinement.

We have now totally reworked this figure to make it more clear. Genomes are now represented by genome maps and mutations are shown above and below each genome map, organized by date (Figure RTR4 and main Figure 4). We put the heatmaps from the original figure 4 into the supplement, and can now be found as Fig S6. Thank you for providing feedback on this as it helped us think of better ways to show this data.

Figure RTR4

Evaluation of non-synonymous variants in prevalent wastewater viruses. (A) Genome map of Human Adenovirus 41 (middle) with non-synonymous variants displayed above (Houston, TX) and below (El Paso, TX) according to genome position (X axis) and date (Y axis). **(B)** Like (A) but with Astrovirus MLB1. **(C)** Like (A) but with JC Polyomavirus. **(D)** t-SNE of non-synonymous variant frequency of astrovirus MLB1. **(E)** Like (D) but with JC Polyomavirus. **(F)** Like (D) but with JC Polyomavirus.

- Line 341

The centrifugation parameters should be elucidated.

We have updated the methods to state the centrifugation step is 3374 x g for 10 minutes.

- Regarding the real-time PCR, the manuscript should comply with MIQE guidelines (Minimum Information for Publication of Quantitative Real-Time PCR Experiments; <https://doi.org/10.1373/clinchem.2008.112797>).

The RT-qPCR method is now modified to comply with MIQE guidelines, the concentration of primers and probes are included. In addition the sequences for oligonucleotide standards used in the experiments are also provided.

- Line 380

The term 'subjective opinion' is unclear in this context.

While it would be challenging to make objective classifications of public health threat of every virus, this list was generated by co-authors on the Texas Epidemic Public Health Institute which includes virologists as well as public health professionals.

- Line 390

Why is the 'quality score < 23' selected?

We only wanted to map high quality reads to avoid false positive mappings.

- Line 398

Why do you use 'either 1000 nucleotides or 50% of the genome/segment length'?

We aimed to be as specific as possible for our measurements, rather than as sensitive as possible. Statistically, this threshold allows very specific detection of viruses.

- Line 432

Is 'RT-PCR' used in this context, or is it RT-qPCR/Real-time RT-PCR?

It is RT-qPCR.

- Line 434

The verb 'was averaged' should be 'were averaged' to agree with the plural subject.

Thank you. We have now fixed this.

- Table 1

The direction of the sequences (e.g., 5'-3') should be indicated in this table.

The direction has been included in the table.

Reviewer #2 (Remarks to the Author):

Wastewater-based epidemiology (WBE) is an upcoming field, which contributed significantly to the understanding of virus transmission during the pandemic. Several studies have used viral titers in wastewater to predict the trajectory of the pandemic. WBE is thus an important tool, which needs to be refined for future applications in public health.

This study is a good example of applied WBE. The authors have refined the technique by selectively enriching clinically relevant human viruses. The technique allowed a deeper insight into the diversity of human viruses in wastewater.

The spatiotemporal distribution of viruses is clearly evident from the results. In the future, it will be interesting to elucidate the factors responsible for this distribution.

Overall the study is an important contribution to this field, which would improve our understanding of viral transmission and its effect on human health.

Minor revisions: 1. In Fig.2, the legend for panel E is missing.

Thank you. We have now fixed this mistake.

2. In the discussion, a section explaining why the numbers are underestimated using qPCR compared to virome sequencing may be included. This is most likely because PCR positivity represents only symptomatic cases. Also not every symptomatic case is tested for qPCR positivity. On the other hand, WBE enables the detection of viruses shed by a large population of asymptomatic cases.

We thank Reviewer #2 for their remarks on this manuscript. Interestingly, we had text about this in an earlier draft of the manuscript but decided to take it out for the sake of being concise. On your recommendation we have added discussion of this phenomenon back into the manuscript (lines 343 – 347).

In the case of SARS-CoV-2 testing, we believe we are observing changes in human behavior around RT-qPCR testing (e.g. fewer individuals getting tested when having upper respiratory symptoms, replacement of PCR testing with rapid antigen testing). However, an increase in asymptomatic infections seems a plausible contributor as well, considering the increase in hybrid immunity to SARS-CoV-2.

REVIEWERS' COMMENTS

Reviewer #1 (Remarks to the Author):

My questions and comments have been addressed by the authors.

Reviewer #2 (Remarks to the Author):

The authors have reasonably addressed all the suggested revisions.